# Pursuing Better Decision Boundaries for Long-Tailed Object Detection via Category Information Amount

**Yanbiao Ma, Wei Dai & Jiayi Chen**
Xidian University, Xi'an, 710071, China
ybmamail@stu.xidian.edu.cn

## Abstract

In object detection, the instance count is typically used to define whether a dataset exhibits a long-tail distribution, implicitly assuming that models will underperform on categories with fewer instances. This assumption has led to extensive research on category bias in datasets with imbalanced instance counts. However, models still exhibit category bias even in datasets where instance counts are relatively balanced, clearly indicating that instance count alone cannot explain this phenomenon. In this work, we first introduce the concept and measurement of category information amount. We observe a significant negative correlation between category information amount and accuracy, suggesting that category information amount more accurately reflects the learning difficulty of a category. Based on this observation, we propose Information Amount-Guided Angular Margin (IGAM) Loss. The core idea of IGAM is to dynamically adjust the decision space of each category based on its information amount, thereby reducing category bias in long-tail datasets. IGAM Loss not only performs well on long-tailed benchmark datasets such as LVIS v1.0 and COCO-LT but also shows significant improvement for underrepresented categories in the non-long-tailed dataset Pascal VOC. Comprehensive experiments demonstrate the potential of category information amount as a tool and the generality of our proposed method.

## 1 Introduction

In object detection tasks, long-tailed distribution is a common phenomenon, where most instances are concentrated in a few categories, while other categories have relatively few instances Jiao et al. (2019); Liu et al. (2020); Zou et al. (2023); Oksuz et al. (2020). A widely accepted perspective is that the imbalance in the number of instances causes the model to be more biased towards frequent categories during training, ignoring the less frequent ones, leading to significant category bias during testing Cho & Krähenbühl (2023); Alshammari et al. (2022); Ren et al. (2020); Cui et al. (2019); Wang et al. (2020a). However, recent research in image classification suggests that category bias is not only caused by the imbalance in sample numbers but may also be closely related to the complexity of intra-category features Ma et al. (2023a;c); Kaushik et al. (2024). This is evidenced in datasets with perfectly balanced samples, where models still exhibit bias. Out of curiosity, we examined the correlation between category average precision (AP) and the number of instances on Pascal VOC, a target detection dataset with a relatively balanced number of instances (see Figure 1), and found that the correlation between the two was very low. This indicates that

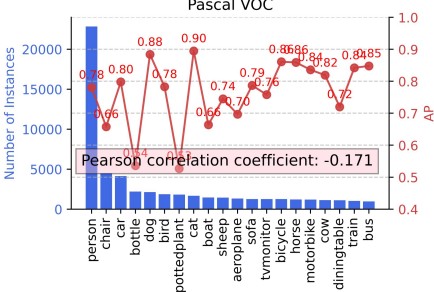

Figure 1: The left vertical axis represents the number of instances per class. The right vertical axis represents the performance of Faster R-CNN trained with cross-entropy loss using R-50-FPN as the backbone across all classes, trained on the Pascal VOC. The model was trained using the settings described in Section 4.2. The red text box displays the Pearson correlation coefficient between class performance and the number of instances.

in object detection, model bias may also originate from the complexity of intra-category features.

Traditional long-tailed object detection methods mainly alleviate this issue by re-weighting low-frequency categories Shen et al. (2016); Gupta et al. (2019); Alshammari et al. (2022); Cui et al. (2019), adjusting gradients Lin et al. (2017b); Wang et al. (2021a); Li et al. (2022); Tan et al. (2020; 2021), and employing data augmentation techniques Ghiasi et al. (2021); Ma et al. (2023b); Zang et al. (2021); Ma et al. (2024b). However, these approaches primarily focus on the impact of the number of instances while ignoring the complexity within categories. As a result, the model may fail to focus on some disadvantaged categories, limiting its overall performance. To better reveal and mitigate category bias, we

Table 1: Pearson correlation coefficient between category information amount and class average precision on long-tailed datasets. The model is Faster R-CNN with R-50-FPN backbone.

| Dataset | LVIS v1.0 | | |
|---------|-----------|--------|-------|
| | CE | SeeSaw | Focal |
| IA | -0.68 | -0.66 | -0.70 |
| Dataset | COCO-LT | | |
| IA | -0.66 | -0.65 | -0.69 |

introduce the concept of category information amount and its corresponding measurement. Category information amount quantifies the diversity of instances within a category, and a straightforward hypothesis is that the greater the information amount of a category, the more difficult it is for the model to learn and memorize that category. We calculated the correlation between category information amount and AP on the relatively balanced Pascal VOC dataset and the long-tailed dataset (LVIS v1.0 and COCO-LT), as shown in Table 1 and Figure 2. Category information amount, compared to the number of instances, better reflects the difficulty of a category. Therefore, we aim to leverage category information amount to direct the model's attention to truly challenging categories rather than simply focusing on categories with fewer instances.

Consider the following scenario: if a head category has very little category information amount, the model can more easily learn and abstract the patterns of that category Cui et al. (2019). From an information compression perspective, the decision space required for that category need not be very large. However, recent studies Wang et al. (2022); Qi et al. (2023) have shown that the severe imbalance in data volume leads to pathological decision boundaries, where the decision space for tail categories is significantly compressed, while head categories have disproportionately large decision spaces. This unreasonable allocation of decision space represents a waste of model capacity. Can we directly equalize decision spaces?

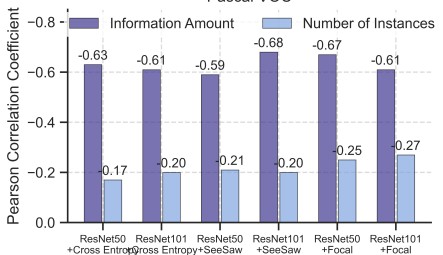

Figure 2: Pearson correlation coefficients between category information amount and category average precision and between category instance count and category average precision, under two backbone networks and three loss function settings.

Differences in category information amount are inherent Ma et al. (2023a); Li & He (2024). For a recognition task, forcibly equalizing decision spaces means that for categories with higher information amounts, the model needs to learn stronger invariant representations to compress such categories into a decision space of the same size as those with lower information amounts. However, the model does not receive additional constraints or support to improve its learning for categories with higher information amounts. Therefore, we propose dynamically adjusting the proportion of decision space allocated to each category based on its information amount. Specifically, we design a novel loss function—Information Amount-Guided Angular Margin (IGAM) Loss, which aims to reduce model bias by dynamically adjusting decision spaces according to the information amount of each category. To enable dynamic updates to the information amount, we also designed a low-cost, end-to-end training strategy. Comprehensive experiments on long-tailed benchmark datasets LVIS v1.0, COCO-LT, and Pascal VOC demonstrate that our method surpasses most existing approaches in both overall performance and reducing model bias.

The main contributions of this paper are as follows: **(1)** We propose the concept of category Information amount and define its measurement (Section 3.1). A surprising finding is that in object detection tasks, category informativeness, compared to the number of instances, better reflects the difficulty of learning each category. This provides a useful tool for future research on improving performance on challenging categories. **(2)** We introduce the Information Amount-Guided Angular Margin (IGAM) Loss (Section 3.2), which adjusts the decision space of categories based on their information amount, encouraging the model to focus more on challenging categories. To dynamically update the information amount of categories, we propose an end-to-end training framework for applying IGAM at a low cost (Section 3.3). **(3)** On long-tail benchmark datasets LVIS v1.0 and

COCO-LT, our method achieves the best performance in most cases, particularly in improving the model's accuracy on rare categories (Sections 4.4 and 4.5). On the relatively balanced Pascal VOC dataset, our method significantly outperforms other approaches on challenging categories.

## 2 RELATED WORK

### 2.1 LONG-TAILED OBJECT DETECTION

In the research of long-tailed object recognition, the main approaches include data re-sampling, specialized loss function design, architectural improvements, decoupled training, and data augmentation. Data re-sampling is a common method to address imbalanced datasets by increasing the sampling frequency of tail class samples to balance the data distribution. Common re-sampling strategies include Class-aware sampling Shen et al. (2016) and Repeat factor sampling (RFS) Gupta et al. (2019). These methods can be employed at different stages of training to achieve a multi-stage training process. Specialized loss function design is another technical approach to tackling long-tailed challenges. For instance, EQL Tan et al. (2020) reduces suppression on tail classes by truncating the negative gradients from head classes. The subsequent EQLv2 Tan et al. (2021) further improves this approach through a gradient balancing mechanism. Other methods, such as Seesaw Loss Wang et al. (2021a), Equalized Focal Loss Li et al. (2022), ACSL Wang et al. (2021b), and LOCE Feng et al. (2021), reduce excessive suppression of tail classes by dynamically adjusting classification logits or suppressing overconfident scores. C2AM Wang et al. (2022) observed that the severe imbalance in weight norms across classes leads to pathological decision boundaries, and therefore proposes learning fairer decision boundaries by adjusting the ratio of weight norms.

Current research mainly focuses on these two directions. In addition, module improvement emphasizes modifying the structure of detectors to address long-tailed distribution issues. For example, BAGS Li et al. (2020) and Forest R-CNN Wu et al. (2020) mitigate the impact of head classes on tail classes by grouping all classes based on valuable prior knowledge. Decoupled training Kang et al. (2019) has found that long-tailed distributions do not significantly affect the learning of high-quality features, thus some methods freeze the feature extractor parameters during the classifier learning phase, adjusting only the classifier Ma et al. (2023b); Wang et al. (2020a); Zhang et al. (2021). Data augmentation, as a means of introducing additional sample variability, has been shown to provide further improvements in long-tailed detection tasks. Recently proposed methods such as Simple Copy-Paste Ghiasi et al. (2021), FDC Ma et al. (2023b), FASA Zang et al. (2021), and FUR Ma et al. (2024b) supplement the insufficiency of tail-class samples by performing data augmentation in both image and feature spaces. RichSem Meng et al. (2024) and Step-wise Learning Dong et al. (2023) introduce Transformer-based object detection architectures, with the former relying on external data and adding new network branches, while the latter incorporates multiple modules and multi-stage training. The core advantage of our proposed IGAM lies in its simplicity and efficiency.

### 2.2 METHODS FOR MEASURING CLASS DIFFICULTY

**The study of class difficulty is most relevant to our work.** Most research addressing class bias has focused on scenarios with sample imbalance, where rebalancing strategies based on sample size can be somewhat effective. However, recent studies have reported that even when sample sizes are perfectly balanced, classification models still exhibit significant performance disparities across different classes. Investigating the root causes of model bias in scenarios where sample sizes are balanced is crucial for improving model fairness and understanding learning mechanisms. However, research on this issue is still limited. From a geometric perspective, DSB Ma et al. (2023a), CR Ma et al. (2023c), and IDR Ma et al. (2024a) conceptualize the data classification process as the disentangling and separating of different perceptual manifolds. These three studies respectively reveal that the geometric properties of perceptual manifolds—volume, curvature, and intrinsic dimensionality—are significantly correlated with class performance. Kaushik et al. (2024) discovered that differences in the spectral features of classes could be a source of class bias. Unfortunately, in the field of object detection, there has been no research exploring the underlying causes of model bias. Our work is the first to directly report on the widespread bias present in object detection models and to attempt to explore the potential mechanisms underlying this bias.

## 3 PURSUING BETTER DECISION BOUNDARIES WITH THE HELP OF CATEGORY INFORMATION AMOUNT

In this section, we first define and compute the category information amount (Section 3.1), then gradually derive how to dynamically adjust the decision space of categories based on their informa-

tion amount (Section 3.2). Finally, we propose a low-cost, end-to-end training strategy to enable the dynamic update of the category information amount (Section 3.3).

## 3.1 DEFINITION AND MEASUREMENT OF CATEGORY INFORMATION AMOUNT

Recent studies have shown that the response of deep neural networks to images is similar to human vision, following the manifold distribution hypothesis, where the embeddings of images lie near a low-dimensional perceptual manifold embedded in high-dimensional space Cohen et al. (2020); Li et al. (2024). Continuous sampling along a dimension of this manifold corresponds to continuous changes in physical features. Therefore, the volume of the perceptual manifold mapped by a deep neural network can effectively measure the information amount of a category. Based on this theory, we define the information amount $I_i$ of category $i$ as the volume of its perceptual manifold: $I_i = Vol(X_i)$, where $X_i = [x_1, x_2, \ldots, x_m]$ represents the set of embeddings for instances in category $i$, $m$ denotes the number of instances. $\text{Vol}(X_i)$ measures the volume of the perceptual manifold, reflecting the information amount of the category. It is important to note that the embeddings used to calculate the information amount should be extracted from the classification module of the object detection model, not the regression module. Below is the method for calculating $\text{Vol}(X_i)$.

Given the embedding set $X_i = [x_1, x_2, \ldots, x_m] \in \mathbb{R}^{p \times m}$, where each instance embedding $x_j \in \mathbb{R}^p$, $p$ denotes the dimension of the embedding. We first compute the covariance matrix of the embedding set $X_i$: $\Sigma(X_i) = \frac{1}{m}\sum_{j=1}^{m}(x_j - \bar{x})(x_j - \bar{x})^T$, where $\bar{x}$ is the mean vector of the embedding set $X_i$: $\bar{x} = \frac{1}{m}\sum_{j=1}^{m} x_j$.

The covariance matrix $\Sigma(X_i)$ captures the distribution characteristics of the category $i$ in the high-dimensional embedding space, and its determinant can be used to calculate the volume of the perceptual manifold. To enhance the accuracy of the covariance matrix estimation, we employ the Ledoit-Péché nonlinear shrinkage method Burda & Jarosz (2022), which improves robustness in high-dimensional spaces through eigenvalue transformation. The covariance matrix $\Sigma(X_i)$ is reconstructed as follows:

Listing 1: Category Information Amount

```
import numpy as np
# Function to compute Information Amount.
def compute_instance_diversity(embeddings):
    m, p = embeddings.shape
    mean_embedding=np.mean(embeddings,axis=0)
    centered_embeddings = embeddings -
        mean_embedding
    sample_cov = np.dot(centered_embeddings.T,
        centered_embeddings) / m
    eigvals, eigvecs = np.linalg.eigh(
        sample_cov)
    c = p / m
    lambda_minus = (1 - np.sqrt(c))**2
    lambda_plus = (1 + np.sqrt(c))**2
    d = np.maximum(eigvals, lambda_minus)
    shrunk_cov = np.dot(eigvecs, np.dot(np.
        diag(d), eigvecs.T))
    Information = 0.5 * np.log2(np.linalg.det(
        np.eye(p) + shrunk_cov))
    return Information
```

$$\Sigma(X_i) = V\text{diag}(\lambda_1, \lambda_2, \ldots, \lambda_p)V^T, \tag{1}$$

where $V$ is the matrix of eigenvectors, and $\lambda_i = \max(\lambda_i, \lambda_-)$, with $\lambda_- = (1 - \sqrt{p/m})^2$ as the nonlinearly transformed minimum eigenvalue. Finally, the information amount of category $i$ is formally defined as:

$$I_i = \text{Vol}(X_i) = \frac{1}{2}\log_2 \det(\Sigma(X_i) + I), \tag{2}$$

where $\det(\Sigma(X_i) + I)$ is the determinant of the matrix $\Sigma(X_i) + I$, and $I$ is the identity matrix. The determinant reflects the spread of the category's embeddings, representing the volume of the perceptual manifold. In this way, we can quantify the information amount of category $i$.

## 3.2 INFORMATION AMOUNT-GUIDED ANGULAR MARGIN (IGAM) LOSS

Cross-entropy loss is widely used in deep learning, especially in classification tasks. It measures the difference between the model's predicted probability distribution and the true label distribution. For the classification component in object detection tasks, given a feature vector $x$ and a label $i$, the cross-entropy loss is typically defined as:

$$L = -\log\left(\frac{e^{W_i^T x}}{\sum_{j=1}^{C} e^{W_j^T x}}\right), \tag{3}$$

where $W_j$ is the $j$-th column of the final fully connected layer, corresponding to the weight vector for category $j$. In long-tailed scenarios, it has been observed that the norm of the weight vectors corresponding to each category is extremely unbalanced, making it more difficult to recognize tail

categories. For example, in a binary classification task, when $W_1^T x = W_2^T x$, we have:

$$\|W_1\|_2 \cdot \cos(\theta_1) = \|W_2\|_2 \cdot \cos(\theta_2), \quad 0 \leq \theta_1, \theta_2 \leq \frac{\pi}{2}. \tag{4}$$

If $\|W_1\|_2 > \|W_2\|_2$, then $\theta_1 > \theta_2$ must hold for the Equation (4) to be true. Clearly, when $\|W_1\|_2 \gg \|W_2\|_2$, the decision space for category 2 is pathologically compressed. To address this, a simple solution is to directly equalize the decision space, ignoring the norm of the weight vectors for each category, resulting in:

$$L = -\log\left(\frac{e^{s \cdot \cos(\theta_i)}}{\sum_{j=1}^{C} e^{s \cdot \cos(\theta_j)}}\right), \tag{5}$$

where $\cos(\theta_i) = \frac{W_i^T x}{\|W_i\|_2 \cdot \|x\|_2}$, and $s$ is a hyperparameter introduced to stabilize training. The optimization goal can be understood as minimizing the angle between $W_i$ and $x$. Although this approach addresses the pathological decision space allocation, recent studies show that absolute equal allocation among all categories is not the optimal solution. A straightforward explanation is that even with the same number of samples for each category, their learning difficulties vary. Thus, absolute equality restricts the model's sensitivity and attention to different categories.

From an information compression perspective, if the information amount of a category is significantly larger than others, but its decision space is required to be compressed to the same extent, this is clearly unreasonable. Therefore, we propose using each category's information amount to dynamically adjust the decision boundaries, allowing categories with larger information amounts to have larger decision spaces. Specifically, we introduce an angular margin $m_{ij}$ based on the information amount of each category into Equation (5). The final optimization objective is expressed as:

$$L = -\log\left(\frac{e^{s \cdot \cos(\theta_i)}}{e^{s \cdot \cos(\theta_i)} + \sum_{j=1,j \neq i}^{C} e^{s \cdot \cos(\theta_j + m_{ij})}}\right), \tag{6}$$

where: $m_{ij} = \max\left(0, \frac{1}{\pi} \cdot \log(\frac{I_i'}{I_j'})\right)$, and: $I_i' = \frac{e^{I_i/(\bar{I} \cdot \sqrt{C})}}{\sum_{j=1}^{C} e^{I_j/(\bar{I} \cdot \sqrt{C})}} \cdot C + 1$, $\bar{I} = \sum_{i=1}^{C} I_i$. In this formula, $I_i'$ represents the normalized information amount of category $i$, with the normalization method adopted from Ma et al. (2023a). The term $m_{ij}$ is based on the ratio of the information amounts of the category $i$ and $j$. If the information amount of category $i$ is larger (i.e., $I_i' > I_j'$), then $m_{ij}$ is positive, meaning the decision space for that category should be expanded. Conversely, if $I_i' < I_j'$, then $m_{ij}$ is negative, and the decision space for class $i$ is compressed. By incorporating information amount, IGAM Loss can more accurately reflect the internal complexity of categories, rather than solely relying on instance numbers. This allows the model to allocate more decision space to complex categories, improving detection accuracy for these categories.

In practical training, we face an engineering challenge: the information amount of categories changes as the model parameters evolve, necessitating dynamic updates. However, calculating the information amount requires the covariance matrix of all instance embeddings, and extracting embeddings for the entire dataset in each iteration would lead to excessive memory and time costs, interrupting training. We propose a novel training framework to address this issue.

### 3.3 LOW-COST DYNAMIC UPDATE OF INFORMATION DENSITY

#### 3.3.1 DYNAMIC UPDATE STRATEGY

The phenomenon of **feature slow drift** Wang et al. (2020b); Ma et al. (2023a) indicates that as training progresses, the distance between the embeddings of the same sample at different training stages becomes increasingly smaller, to the extent that the previous version of an embedding can approximate the latest version. Inspired by this, we propose the most straightforward approach: store the embeddings of all instances generated during training in a queue, with the queue length equal to the total number of instances in the dataset. After each training epoch, all embeddings in the queue can be updated once. At the end of each epoch, the embeddings in the queue are used to calculate and update the information amount for all categories. In the following, we refer to this approach as the original strategy. Although the original strategy avoids repeatedly extracting embeddings for the entire dataset, it increases the demand for storage space. Considering that the essence of calculating the information amount lies in obtaining the covariance matrix of the instance embeddings, we propose a new strategy that significantly reduces storage space. The core idea of this new strategy is to calculate the global covariance matrix of the samples using multiple local sample covariance matrices. The specific steps are as follows:

(1) Initialize a queue to store instance embeddings. The length of the queue can be adjusted according to the GPU memory size. Suppose the object detection dataset contains $C$ categories with a total of $N$ instances, and the queue length is $d$. If $d < N$, it means the queue cannot hold all instance embeddings. In this work, we set the queue length to 50,000, which can store 50,000 instance embeddings.

(2) At the beginning of a training epoch, first store the instance embeddings generated in each batch into the queue until it is full (i.e., storing $d$ instance embeddings). Then, use the embeddings in the queue to calculate the local covariance matrix and mean for each category. Continuously update the queue, and once all old embeddings in the queue are updated, calculate the local covariance matrix and mean for each category again. By the end of an epoch, we can calculate $\lfloor N/d \rfloor + 1$ local sample covariance matrices $\Sigma_i^k$ and means $\mu_i^k$ for each category $i = 1, \ldots, C$, $k = 1, \ldots, \lfloor N/d \rfloor + 1$.

(3) At the end of an epoch, use the stored local covariance matrices to calculate and update the information amount for each category. Taking category $i$ as an example, first calculate the global mean:

$$\mu_i = \frac{1}{N_i} \sum_{k=1}^{\lfloor N/d \rfloor + 1} n_i^k \mu_i^k, \tag{7}$$

where $N_i$ is the total number of instances in category $i$, and $n_i^k$ is the number of instances in the local sample. Then, calculate the global covariance matrix:

$$\Sigma_i = \frac{1}{N_i} \left( \sum_{k=1}^{\lfloor N/d \rfloor + 1} n_i^k \Sigma_i^k + \sum_{k=1}^{\lfloor N/d \rfloor + 1} n_i^k (\mu_i^k - \mu_i)(\mu_i^k - \mu_i)^T \right). \tag{8}$$

The proof of this formula is provided in Appendix A. By integrating local covariance matrices to obtain the global covariance matrix, we significantly reduce the additional storage space required to update the information amount. Further, the information amount of category $i$ is estimated as $\text{Vol}_i = \frac{1}{2} \log_2 \det(I + \Sigma_i)$.

### 3.3.2 STORAGE SPACE COMPARISON

Assume the object detection dataset contains $N$ instances, each with an embedding dimension of $p$, and there are $C$ categories. The queue length is set to $d$. The storage space required by the original strategy is: $S_{\text{original}} = N \times p$. The storage space required by the new strategy is:

$$S_{\text{new}} = d \times p + C \times (\lfloor N/d \rfloor + 1) \times p^2. \tag{9}$$

where $C \times (\lfloor N/d \rfloor + 1) \times p^2$ represents the space needed to store the local covariance matrices. To analyze when the new strategy saves more space, we define the storage space ratio $R$:

$$R = \frac{S_{\text{new}}}{S_{\text{original}}} = \frac{d \times p + C \times (\lfloor N/d \rfloor + 1) \times p}{N}. \tag{10}$$

When $R < 1$, the new strategy saves more storage space. To visually compare the storage space requirements of the new and original strategies, we take the Pascal VOC and MS COCO datasets as examples and plot the function graph of the storage space ratio $R$ as it varies with the queue length $d$ in Figure 3. It can be observed that in most cases, the storage space required by the new strategy is less than that of the original strategy. By visualizing our proposed storage space ratio, it becomes easy to choose the optimal queue length $d$, thereby saving approximately 60% of the stor-

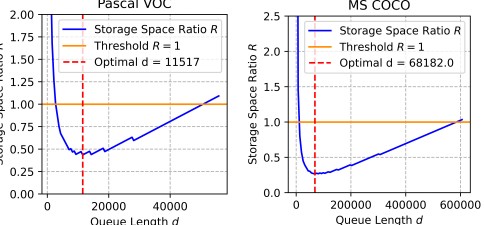

Figure 3: The function of the storage space ratio $R$ as it varies with the queue length $d$ on the Pascal VOC and MS COCO datasets.

age space. Given two examples $\{N = 55800, p = 128, C = 20\}$ and $\{N = 605638, p = 128, C = 80\}$, corresponding to the Pascal VOC and MS COCO datasets respectively, we use Listing 2 to select the optimal queue lengths $d$ of 11517 and 68182. For Example 1, the original strategy requires an additional 27.25 MB of memory, while the new strategy only requires 11.87 MB, saving approximately **56.44%** of memory. For Example 2, the new strategy can reduce memory usage from 295.72 MB to 78.29 MB, saving approximately **73.52%** of memory.

The new training framework significantly reduces storage space utilization by merging the local sample covariance matrices, ensuring that the calculated value of information density remains unchanged. This innovative strategy not only provides an efficient solution for dynamically updating information density but also offers a low-cost storage solution for future research.

Listing 2: Code for selecting the optimal $d$

```python
import matplotlib.pyplot as plt
import numpy as np
# Define parameters
N = 55800  # Total number of instances
p = 128      # Embedding dimension
C = 20       # Number of categories
# Define the range of queue lengths
d_values = np.linspace(1000, N, 100)
R_values = list((d_values + C * (np.floor(N /
    d_values) + 1) * p) / N)
min_R_values = min(np.abs(R_values))
optimal_d = d_values[R_values.index(
    min_R_values)]
print('The optimal d is:', optimal_d)
```

## 4 EXPERIMENTS

We conducted a comprehensive evaluation of the effectiveness of IGAM on long-tailed and relatively balanced object detection benchmark datasets. The experiments are divided into two parts: the first part is carried out on the long-tailed large-vocabulary datasets LVIS v1.0 and COCO-LT, while the second part is conducted on the relatively balanced Pascal VOC dataset.

### 4.1 DATASETS AND EVALUATION METRICS

**LVIS v1.0** Gupta et al. (2019) contains 1,203 categories, with the training set consisting of 100k images (approximately 1.3M instances) and the validation set containing 19.8k images. Based on the frequency of occurrence in the training set, all categories are divided into three groups: rare (1~10 images), common (11~100 images), and frequent (more than 100 images). In line with EFL Li et al. (2022), we report not only the widely used object detection metric $AP^b$ across IOU thresholds (from 0.5 to 0.95) but also the bounding box AP for frequent ($AP_f$), common ($AP_c$), and rare ($AP_r$) categories separately. The **COCO-LT** Wang et al. (2020a) dataset is a long-tailed subset of MS COCO Lin et al. (2014), and they share the same validation set. Consistent with previous work Wang et al. (2021a), we divided the 80 classes in COCO-LT into four groups based on the number of training instances per class: fewer than 20 images, 20~400 images, 400~8000 images, and 8000 or more images. **Pascal VOC** Everingham et al. (2015) includes two versions, 2007 and 2012, comprising a total of 20 classes. Following standard practice Tong & Wu (2023), we trained on the train+val sets of VOC2007 and VOC2012 and tested on the test set of VOC2007. We report the *Average Precision (AP)* for each class on Pascal VOC, and on COCO-LT, we report the accuracy of the four groups as $AP_1^b$, $AP_2^b$, $AP_3^b$, and $AP_4^b$. The mean average precision is reported as $mAP^b$.

### 4.2 IMPLEMENTATION DETAILS

Consistent with previous studies Qi et al. (2023), we implemented the Faster R-CNN Ren et al. (2015) detector using the MMDetection Chen et al. (2019) toolbox and adopted ResNet-50 and ResNet-101 He et al. (2016) with an FPN Lin et al. (2017a) structure as the backbone networks. During training, we set the batch size to 16 and the initial learning rate to 0.02, consistent with EFL Li et al. (2022) and C2AM Wang et al. (2022). We trained the model using an SGD optimizer with a momentum of 0.9 and a weight decay rate of 0.0001 for 24 epochs. The learning rate was reduced to 0.002 and 0.0002 at the end of the 16th and 22nd epochs, respectively. In all experiments, we applied random horizontal flipping and multi-scale jittering for data augmentation. We did not use any test-time augmentations. We did not use any test-time augmentations.

### 4.3 MAIN RESULTS: EFFECTIVENESS OF IGAM LOSS

Recalling Section 3.2, we introduce the hyperparameter $s$ in the IGAM Loss, which scales the cosine value to ensure it falls within an appropriate range, thereby stabilizing the optimization process. This is a standard practice in cosine-based classifiers and has been widely adopted. We refer to Wang et al. (2022) for rigorous determination of $s$ and tested the model's performance under different settings. The experimental results, summarized in Table 2, show that when $s = 30$, the model trained with IGAM Loss achieves optimal performance. After determining the hyperparameter, we

Table 2: Results of IGAM Loss with different hyper-parameter $s$ on LVIS v1.0. The model is Faster R-CNN with R-50-FPN backbone.

| s | $mAP^b$ | $AP_r$ | $AP_c$ | $AP_f$ |
|---|---------|--------|--------|--------|
| 10 | 16.8 | 0.8 | 12.6 | 27.8 |
| 20 | 26.2 | 17.6 | 23.8 | 31.5 |
| 30 | **26.8** | 19.0 | 25.2 | 31.4 |
| 40 | 25.9 | 18.4 | 23.5 | 30.9 |
| 50 | 24.6 | 16.9 | 22.7 | 30.5 |

Table 3: Evaluation results on LVIS v1.0. The $mAP^b$, $AP_r$, $AP_c$, and $AP_f$ (%) for each method are reported, with green arrows indicating performance improvements.

| Framework | Backbone | Loss | $mAP^b$ | $AP_r$ | $AP_c$ | $AP_f$ |
|---|---|---|---|---|---|---|
| Faster R-CNN | ResNet-50-FPN | Cross-Entropy (CE) | 19.3 | 1.1 | 16.1 | 30.9 |
| | | IGAM Loss | 26.8 ↑7.5 | 19.0 ↑17.9 | 25.2 | 31.4 |
| | ResNet-101-FPN | Cross-Entropy (CE) | 20.9 | 1.0 | 18.2 | 32.7 |
| | | IGAM Loss | 28.0 ↑7.1 | 20.1 ↑19.1 | 26.8 | 32.5 |
| | Swin-T | Cross-Entropy (CE) | 25.4 | 6.2 | 24.5 | 35.3 |
| | | IGAM Loss | 31.7 ↑6.3 | 21.4 ↑15.2 | 30.8 | 37.1 |
| Cascade Mask R-CNN | ResNet-50-FPN | Cross-Entropy (CE) | 22.7 | 1.5 | 20.6 | 34.4 |
| | | IGAM Loss | 29.1 ↑6.4 | 21.5 ↑20.0 | 27.7 | 33.9 |
| | ResNet-101-FPN | Cross-Entropy (CE-FPN) | 24.5 | 2.6 | 23.1 | 35.8 |
| | | IGAM Loss | 29.7 ↑5.2 | 21.9 ↑19.3 | 28.5 | 34.6 |
| | Swin-T | Cross-Entropy (CE) | 31.3 | 6.8 | 30.2 | 39.4 |
| | | IGAM Loss | 37.9 ↑6.6 | 25.2 ↑18.4 | 35.5 | 38.7 |
| DETR | ResNet-50-FPN | Cross-Entropy (CE) | 21.8 | 3.3 | 21.2 | 30.5 |
| | | IGAM Loss | 27.6 ↑5.8 | 18.5 ↑15.2 | 27.0 | 32.7 |
| | ResNet-101-FPN | Cross-Entropy (CE) | 23.1 | 3.7 | 23.4 | 32.2 |
| | | IGAM Loss | 30.4 ↑7.3 | 20.7 ↑17.0 | 30.0 | 35.5 |
| | Swin-T | Cross-Entropy (CE) | 30.2 | 6.3 | 28.9 | 38.2 |
| | | RichSem Meng et al. (2024) | 34.9 | 26.0 | 32.6 | 41.3 |
| | | IGAM Loss | 37.3 ↑7.1 | 24.8 ↑18.5 | 34.8 | 38.3 |

Table 4: Performance comparison with state-of-the-art methods on LVIS *val* set. The ResNet-50-FPN and ResNet-101-FPN are adopted as backbones for Faster R-CNN. All methods are trained with a 2x schedule, *i.e.*, 24 epochs in total. The $mAP^b$, $AP_r$, $AP_c$, and $AP_f$ (%) for each method are reported. The best and second-best results are shown in **underlined bold** and **bold**, respectively.

| Strategy | Methods | LVIS v1.0 | | | | | | | |
|---|---|---|---|---|---|---|---|---|---|
| | | ResNet-50-FPN | | | | ResNet-101-FPN | | | |
| | | $mAP^b$ | $AP_r$ | $AP_c$ | $AP_f$ | $mAP^b$ | $AP_r$ | $AP_c$ | $AP_f$ |
| End-to-end | RFS Gupta et al. (2019) | 24.2 | 14.2 | 22.3 | 30.6 | 25.7 | 15.9 | 23.7 | 32.2 |
| | EQL Tan et al. (2020) | 21.8 | 3.6 | 21.1 | 30.5 | 23.4 | 4.5 | 22.9 | 30.3 |
| | DropLoss Hsieh et al. (2021) | 21.8 | 5.2 | 21.8 | 29.1 | 23.5 | 5.9 | 23.9 | 30.7 |
| | RIO Chang et al. (2021) | 23.4 | 15.3 | 21.2 | 29.4 | 25.5 | 17.2 | 23.7 | 31.2 |
| | Forest R-CNN Wu et al. (2020) | - | - | - | - | - | - | - | - |
| | BALMS Ren et al. (2020) | 24.1 | 15.2 | 23.0 | 29.4 | 26.9 | 18.5 | 25.2 | 32.4 |
| | De-confound-TDE Tang et al. (2020) | 23.7 | 10.0 | 22.4 | 31.2 | - | - | - | - |
| | EQLv2 Tan et al. (2021) | 25.4 | 15.8 | 23.5 | **31.7** | 26.8 | 17.1 | 24.9 | **33.1** |
| | Seesaw Wang et al. (2021a) | 24.8 | 14.8 | 22.7 | **31.6** | 26.6 | 14.9 | 25.2 | **33.3** |
| | FASA Zang et al. (2021) | 21.5 | 7.4 | 19.2 | 30.2 | 22.9 | 9.0 | 20.6 | 31.6 |
| | EFL Li et al. (2022) | 26.0 | **16.6** | 25.1 | 30.8 | 26.3 | **18.5** | 23.9 | 32.6 |
| | C2AM Wang et al. (2022) | 25.4 | 15.6 | 24.2 | 30.9 | 26.5 | 18.1 | 25.5 | 31.2 |
| Decoupled | SimCal Wang et al. (2020a) | - | - | - | - | - | - | - | - |
| | BAGS Li et al. (2020) | 23.7 | 14.2 | 22.2 | 29.6 | 25.4 | 14.9 | 25.2 | 31.4 |
| | ACSL Wang et al. (2021b) | 22.2 | 9.9 | 21.3 | 28.5 | 23.7 | 11.0 | 23.0 | 30.2 |
| | DisAlign Zhang et al. (2021) | 20.9 | 3.9 | 20.4 | 29.0 | 25.5 | 13.3 | 24.5 | 32.0 |
| | LOCE Feng et al. (2021) | 25.1 | 15.7 | 24.2 | 30.1 | 26.7 | 18.4 | 25.5 | 31.7 |
| | BACL Qi et al. (2023) | **26.1** | 16.0 | **25.7** | 30.9 | **27.8** | 18.1 | **27.3** | 32.6 |
| End-to-end | IGAM | **26.8** | **19.0** | 25.2 | 31.4 | **28.0** | **20.1** | 26.8 | 32.5 |

compare IGAM Loss with the baseline model. Since our method is derived from cross-entropy loss, the baseline model is trained using cross-entropy loss.

To validate the effectiveness of IGAM Loss, we employed Faster R-CNN and Cascade Faster R-CNN as detection frameworks, with ResNet-50 and ResNet-101 as backbone networks. The baseline models were trained using cross-entropy loss. As shown in Table 3, the four baseline models trained with cross-entropy loss achieve an average precision (APr) of only 1.1%, 1.0%, 1.5%, and 2.6% on tail classes, making it nearly impossible to recognize tail instances. In contrast, IGAM significantly improves the detection accuracy for tail classes. For example, with Mask R-CNN as the framework and ResNet-50-FPN as the backbone, IGAM raises APr by **17.9%**. Moreover, IGAM also brings a **9.1%** performance gain in the APc metric. Beyond the substantial improvements in tail-class performance, IGAM surpasses the baseline model in overall performance across the four different detection frameworks and backbone configurations by **7.5%**, **7.1%**, **6.4%**, and **5.2%**, respectively.

IGAM demonstrates remarkable generalization capabilities across various configurations, and its performance leap over the baseline models highlights the effectiveness and versatility of our proposed method, which refines decision space partitioning dynamically using information amount.

## 4.4 COMPARISON WITH STATE-OF-THE-ARTS

Table 4 presents the experimental results on LVIS v1.0. IGAM outperforms the current state-of-the-art methods on both ResNet-50-FPN and ResNet-101-FPN backbones, achieving overall performances of **26.8%** and **28.0%**, respectively. Notably, for rare categories, IGAM surpasses the second-best method by **2.4%** and **1.6%** on the two backbones, respectively. It is worth mentioning that despite BACL incorporating a series of techniques, including foreground-balanced loss, synthetic hallucination samples, and decoupled training, our method still exhibits strong competitiveness. For the most frequent categories, although EQLv2 and Seesaw exhibit exceptional performance, they significantly lack attention to rare categories. In contrast, IGAM demonstrates superior performance on rare categories while maintaining competitive results on the most frequent categories. On the ResNet-50-FPN backbone, IGAM's $AP_f$ is only **0.3%** and **0.2%** lower than that of EQLv2 and Seesaw, respectively. We attribute IGAM's superior overall performance to its accurate measurement of category learning difficulty through information amount, allowing IGAM not to impair the performance of frequent categories by focusing too much on rare categories.

Table 5: Evaluation results on COCO-LT. The $mAP^b$, $AP_1^b$, $AP_2^b$, $AP_3^b$, and $AP_4^b$ (%) for each method are reported. An asterisk (*) indicates results reproduced by our implementation. The best and second-best results are shown in **underlined bold** and **bold**, respectively.

| Methods | COCO-LT | | | | | | | | | |
|---|---|---|---|---|---|---|---|---|---|---|
| | ResNet-50-FPN | | | | | ResNet-101-FPN | | | | |
| | $mAP^b$ | $AP_1^b$ | $AP_2^b$ | $AP_3^b$ | $AP_4^b$ | $mAP^b$ | $AP_1^b$ | $AP_2^b$ | $AP_3^b$ | $AP_4^b$ |
| Cross-Entropy (CE) | 24.5 | 0 | 14.6 | **29.6** | **32.9** | 26.0 | 0 | 16.4 | **31.4** | **34.2** |
| Seesaw Wang et al. (2021a) | 23.9 | 3.0 | 14.5 | 28.4 | 32.3 | 24.9 | 3.2 | 14.5 | 30.0 | 33.4 |
| EQLv2 Tan et al. (2021) | **25.7** | 3.8 | **18.1** | 29.6 | **33.0** | **26.8** | 3.2 | **19.4** | 30.8 | 34.1 |
| EFL* Li et al. (2022) | 25.0 | **3.8** | 16.3 | 29.5 | 32.5 | 25.4 | **3.6** | 16.5 | 30.2 | 32.8 |
| C2AM* Wang et al. (2022) | 24.7 | 2.8 | 15.6 | 29.4 | 32.3 | 25.1 | 2.9 | 15.6 | 30.2 | 32.7 |
| IGAM | **25.8** | **6.1** | 18.0 | **29.7** | 32.5 | **27.0** | **6.6** | 19.0 | 30.5 | 33.4 |

## 4.5 EVALUATION RESULTS ON COCO-LT

The COCO-LT dataset is not yet a widely recognized benchmark in the field of long-tailed object detection, and thus, there are relatively few studies validating methods on it. We trained baseline models on COCO-LT using cross-entropy loss and compared Seesaw and EQLv2 following Qi et al. (2023). Additionally, we independently implemented EFL and C2AM. The experimental results are summarized in Table 5. It can be observed that IGAM achieves the best overall performance on both backbone networks, with **25.8%** and **27.0%**, respectively. Notably, IGAM surpasses the second-best method in the $AP_1^b$ metric by **2.3%** and **3.0%** on the two backbones, respectively, highlighting the significant advantage of our method on rare categories. While cross-entropy loss and EQLv2 perform well on the most frequent categories, they exhibit a clear gap in performance on rare categories compared to IGAM. Furthermore, IGAM does not fall behind cross-entropy loss and EQLv2 in the $AP_4^b$ metric, demonstrating that our method effectively balances performance across both rare and frequent categories.

## 4.6 EVALUATION RESULTS ON PASCAL VOC

We trained models using Seesaw, EFL, and C2AM losses on the relatively balanced Pascal VOC dataset for comparison. Table 6 presents the performance of each method across all categories as well as the overall performance. It can be observed that IGAM outperforms the other methods in terms of overall performance on both backbone networks, achieving **77.7%** and **78.6%**, respectively, surpassing the second-best method by 0.8% and 1.1%.

**More importantly**, our method significantly improves the performance of underperforming categories. We selected five representative poorly performing categories for observation: aeroplane, boat, bottle, chair, and pottedplant. With the ResNet-50 backbone, IGAM achieved the best performance across all five categories, surpassing the second-best method by **2.2%**, **1.3%**, **3.4%**, **0.7%**, and **2.1%**, respectively. Similarly, with the ResNet-101 backbone, IGAM also achieved the best performance across all five categories, exceeding the second-best method by **1.6%**, **2.7%**, **3.0%**,

Table 6: Evaluation results on Pascal VOC. $AP^b(\%)$ for each class using different methods with ResNet-50 and ResNet-101 as the backbone. We selected the five most challenging classes, with the best and second-best results are shown in **underlined bold** and **bold**, respectively. Green arrows and values indicate the difference between the best and second-best results.

| Class | ResNet-50 | | | | | | ResNet-101 | | | | | |
|---|---|---|---|---|---|---|---|---|---|---|---|---|
| | CE | Seesaw | EFL | C2AM | BACL | IGAM | CE | Seesaw | EFL | C2AM | BACL | IGAM |
| cat | 84.8 | 89.5 | 88.3 | 88.1 | 89.6 | 87.8 | 88.9 | 91.4 | 91.2 | 89.8 | 91.8 | 88.0 |
| car | 75.7 | 79.8 | 78.9 | 80.8 | 80.3 | 81.3 | 79.2 | 82.2 | 81.1 | 82.0 | 82.7 | 83.3 |
| horse | 81.6 | 85.9 | 86.3 | 87.0 | 85.5 | 88.1 | 83.6 | 86.5 | 86.6 | 87.6 | 86.7 | 87.1 |
| bus | 80.2 | 84.8 | 82.9 | 84.5 | 84.1 | 85.2 | 82.4 | 83.6 | 84.4 | 85.3 | 86.1 | 86.5 |
| bicycle | 82.1 | 86.1 | 84.5 | 85.2 | 87.8 | 84.6 | 84.0 | 87.0 | 85.8 | 85.5 | 88.1 | 86.2 |
| dog | 81.7 | 88.4 | 85.3 | 86.5 | 88.0 | 87.4 | 88.9 | 90.8 | 90.2 | 87.1 | 91.3 | 86.7 |
| person | 75.8 | 79.1 | 77.1 | 77.8 | 80.5 | 79.5 | 81.1 | 84.9 | 82.8 | 78.2 | 85.6 | 79.8 |
| train | 80.4 | 84.2 | 82.9 | 85.3 | 85.6 | 87.0 | 83.9 | 86.4 | 85.4 | 85.5 | 87.7 | 86.1 |
| motorbike | 79.4 | 83.6 | 80.1 | 82.0 | 82.6 | 83.2 | 83.2 | 87.4 | 84.1 | 82.6 | 86.8 | 83.6 |
| cow | 80.3 | 82.9 | 79.8 | 80.7 | 82.8 | 82.1 | 78.4 | 81.8 | 80.1 | 81.0 | 81.5 | 81.9 |
| aeroplane | 64.4 | 69.7 | **72.1** | 71.8 | 70.1 | **74.3** ↑2.2 | 69.0 | **73.1** | 70.0 | 72.5 | 71.3 | **74.7** ↑1.6 |
| tvmonitor | 71.3 | 75.8 | 76.4 | 74.1 | 75.9 | 75.7 | 68.0 | 75.2 | 69.9 | 77.0 | 71.6 | 78.5 |
| sheep | 73.4 | 76.5 | 74.8 | 75.6 | 78.8 | 76.5 | 75.9 | 80.4 | 78.2 | 75.5 | 79.6 | 77.4 |
| bird | 75.5 | 79.3 | 75.2 | 79.7 | 77.3 | 81.0 | 69.5 | 74.1 | 72.3 | 80.4 | 72.7 | 81.2 |
| diningtable | 70.8 | 74.0 | 71.3 | 73.4 | 76.3 | 75.1 | 68.7 | 75.5 | 73.3 | 74.3 | 74.7 | 76.1 |
| sofa | 75.2 | 78.7 | 72.7 | 79.5 | 76.6 | 80.8 | 70.6 | 77.2 | 72.5 | 79.7 | 73.5 | 80.6 |
| boat | 61.8 | **66.4** | 62.4 | 64.2 | 65.2 | **67.7** ↑1.3 | 61.5 | 63.3 | 64.2 | **65.2** | 64.2 | **67.9** ↑2.7 |
| bottle | 50.6 | 53.6 | 50.8 | **54.8** | 54.2 | **58.2** ↑3.4 | 50.0 | 52.4 | 51.5 | **55.6** | 53.6 | **58.6** ↑3.0 |
| chair | 61.5 | **65.8** | 61.1 | 62.3 | 61.7 | **66.5** ↑0.7 | 58.9 | 62.3 | 61.9 | **63.5** | 63.1 | **66.0** ↑2.5 |
| pottedplant | 49.3 | **52.7** | 49.1 | 51.5 | 51.1 | **54.8** ↑2.1 | 48.2 | 53.9 | 50.0 | **54.0** | 52.2 | **57.3** ↑3.3 |
| Total | 72.8 | **76.9** | 74.6 | 76.2 | 76.8 | **77.7** ↑0.8 | 73.5 | **77.5** | 75.8 | 77.0 | 77.3 | **78.6** ↑1.1 |

**2.5%**, and **3.3%**, respectively. Notably, for the two worst-performing categories, bottle, and pottedplant, IGAM consistently demonstrated the most significant improvements. The experimental results indicate that even on relatively balanced datasets, our method can effectively focus on and enhance the performance of the underrepresented categories. This further validates that the category information amount we proposed more accurately measures the learning difficulty of each category.

## 4.7 EFFECTIVENESS IN REDUCING MODEL BIAS

To more clearly demonstrate the effectiveness of our method in mitigating model bias, we use the variance of class-wise average precision (AP) as a measure of model bias. The comparison results on LVIS v1.0 are shown in Figure 4. It can be observed that the models trained with our method exhibit lower bias compared to Seesaw, EFL, and C2AM, across two different backbones. Notably, compared to Seesaw, our method reduces model bias by approximately **50%**. These results are attributed to the accurate reflection of learning difficulty through category information amount. We encourage other researchers to explore additional potential factors influencing model bias, aiming to design more equitable object detection models.

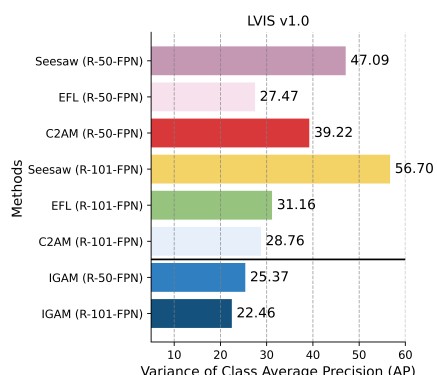

Figure 4: Model bias from models trained with different methods on LVIS v1.0.

## 5 CONCLUSION

This work addresses the issue that instance count fails to explain the generalized bias present in deep learning models for object detection tasks. We propose using information amount to measure the detection difficulty of categories, and experiments reveal a significant negative correlation between a category's information amount and its accuracy. Based on this finding, we propose dynamically adjusting the decision boundaries of categories using their information amount. Comprehensive empirical studies demonstrate that information amount helps the model focus more on learning challenging categories, both in long-tailed and non-long-tailed datasets.

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

## A  APPENDIX

*Proof 1: Integrating Local Covariance Matrices to Obtain the Global Covariance Matrix.* Assume we have a dataset containing $N$ instances, and we divide these instances into $K$ batches, each containing $n_k$ instances. For the $k$-th batch, let the instances be $\{x_{k1}, x_{k2}, \ldots, x_{kn_k}\}$. The mean vector and local covariance matrix for this batch are defined as follows:

$$\mu_k = \frac{1}{n_k} \sum_{i=1}^{n_k} x_{ki},$$

$$\Sigma_k = \frac{1}{n_k} \sum_{i=1}^{n_k} (x_{ki} - \mu_k)(x_{ki} - \mu_k)^T.$$

The global covariance matrix is the covariance matrix of all batches, defined as:

$$\mu = \frac{1}{N} \sum_{k=1}^{K} \sum_{i=1}^{n_k} x_{ki},$$

$$\Sigma = \frac{1}{N} \sum_{k=1}^{K} \sum_{i=1}^{n_k} (x_{ki} - \mu)(x_{ki} - \mu)^T.$$

First, calculate the global mean $\mu$:

$$\mu = \frac{1}{N} \sum_{k=1}^{K} \sum_{i=1}^{n_k} x_{ki} = \frac{1}{N} \sum_{k=1}^{K} n_k \mu_k.$$

Then, split $(x_{ki} - \mu)$ in the global covariance matrix $\Sigma$ into $(x_{ki} - \mu_k)$ and $(\mu_k - \mu)$:

$$\Sigma = \frac{1}{N} \sum_{k=1}^{K} \sum_{i=1}^{n_k} [(x_{ki} - \mu_k + \mu_k - \mu)(x_{ki} - \mu_k + \mu_k - \mu)^T].$$

Expanding this, we get:

$$\Sigma = \frac{1}{N} \sum_{k=1}^{K} \sum_{i=1}^{n_k} [(x_{ki} - \mu_k)(x_{ki} - \mu_k)^T + (x_{ki} - \mu_k)(\mu_k - \mu)^T$$
$$+ (\mu_k - \mu)(x_{ki} - \mu_k)^T + (\mu_k - \mu)(\mu_k - \mu)^T].$$

According to the properties of the covariance matrix, the first term is the local covariance matrix $\Sigma_k$, and the expectation values of the second and third terms are zero. The fourth term can be calculated as:

$$\sum_{i=1}^{n_k} (\mu_k - \mu)(\mu_k - \mu)^T = n_k (\mu_k - \mu)(\mu_k - \mu)^T.$$

Finally, the expression for the global covariance matrix is:

$$\Sigma = \frac{1}{N} \left( \sum_{k=1}^{K} n_k \Sigma_k + \sum_{k=1}^{K} n_k (\mu_k - \mu)(\mu_k - \mu)^T \right).$$

This formula demonstrates that the global covariance matrix can be calculated by taking a weighted sum of the local covariance matrices and adding the difference terms between local means and the global mean. This integration method effectively utilizes the unbiasedness and independence of the local covariance matrices, ensuring the accuracy of the global covariance matrix.  □

## B    SUPPLEMENTARY EXPERIMENTS ON THE PASCAL VOC DATASET

We have also included the improvement brought by IGAM to the baseline methods when using Cascade Mask R-CNN and DETR as target detection frameworks. The experimental results are shown in the Table 7, and it can be observed that IGAM significantly improves the performance of the baseline methods in all four cases.

Table 7: Evaluation results on Pascal VOC.

| Framework | Backbone | Loss | $mAP^b$ |
|---|---|---|---|
| Cascade Mask R-CNN | ResNet-50-FPN | Cross-Entropy (CE) | 74.1 |
|  |  | IGAM Loss | 78.7 |
|  | ResNet-101-FPN | Cross-Entropy (CE) | 75.6 |
|  |  | IGAM Loss | 80.2 |
| DETR | ResNet-50-FPN | Cross-Entropy (CE) | 75.8 |
|  |  | IGAM Loss | 80.5 |
|  | ResNet-101-FPN | Cross-Entropy (CE) | 76.5 |
|  |  | IGAM Loss | 81.0 |

