# OpenReview forum: "Pursuing Better Decision Boundaries for Long-Tailed Object Detection via Category Information Amount"
_ICLR.cc/2025/Conference — ICLR 2025 Poster_

### Official Review · Reviewer_TGj3 · 2024-10-25

**Soundness:** 3
**Presentation:** 4
**Contribution:** 4
**Rating:** 8
**Confidence:** 3

**Summary:**

This paper aims to solve the longtail problem in object detection. The authors find that the instance amount cannot be a good metric to balance the training. They introduce the category information amount and design a new loss function with it to solve the problem.

**Strengths:**

I like this paper. The motivation is clear and the introduction delivers the idea very well. The novelty of this paper is good. The finding and solution are interesting, and straightforward but effective. The experiments can support their method. If all the content is original, then it deserves to be published.

**Weaknesses:**

Some suggestions:
1. Fig.1 please align the class names and the axis ticks.
2. Fig.2 is good to show the correlation between CIA and ACC, but it would be better to show the CIA figure here, or in the ablation study section.
3. explain the symbol $m$ in the Line 170, $p$ in the Line 176.

**Questions:**

1. Do you have any reference to support Line 164-169?
2. Do you maintain several individual queues for different categories, or a single queue for all categories?

---

> ### Author Response · Authors · 2024-11-23
> **Response to Reviewer TGj3.（Weaknesses 1, 2, and 3）**
>
> **Response to Weaknesses 1, 2, and 3:**
>
> We greatly appreciate your recognition of this work. Your careful attention to the scaling issue in Figure 1 demonstrates your rigorous academic pursuit. In line 170, \( m \) represents the number of instances in category \( i \), and in line 176, \( p \) denotes the dimensionality of the instance embeddings. We have added explanations for these two symbols in the revised manuscript.
>
> Regarding your suggestion to present the CIA plot in the ablation study section, we hope you can understand our decision to place it in the introduction. Our goal was to tell a cohesive story, with each step supported by results, which is why we positioned it in the introduction.
>
> Once again, we sincerely wish you a wonderful day.

---

> ### Author Response · Authors · 2024-11-23
> **Response to Reviewer TGj3. （Question 1 and 2）**
>
> **Response to Question 1 and 2:**
>
> In the introduction section, we wrote: "Recent studies have shown that the response of deep neural networks to images is similar to human vision, following the manifold distribution hypothesis, where the embeddings of images lie near a low-dimensional perceptual manifold embedded in high-dimensional space. Continuous sampling along a dimension of this manifold corresponds to continuous changes in physical features." This viewpoint is supported by theory and is based on recent research, which I find fascinating for its connection to neural mechanisms. You may refer to the following papers for further reading: *Separability and Geometry of Object Manifolds in Deep Neural Networks* and *Representations and Generalization in Artificial and Brain Neural Networks*. I believe the theory of disentangling manifolds can be used to further explain some phenomena in deep neural networks, and if you are interested, we can connect after the conference for further discussion and potential collaboration.
>
> **Regarding Question 2**, maintaining a queue for all categories is sufficient. In our code implementation, we appended the category label to each instance embedding, so we only need to retrieve the corresponding category based on the label to calculate the category information amount. Thank you once again for your recognition of our work.

---

> ### Comment · Reviewer_TGj3 · 2024-11-24
>
> I will keep my rating.

---

### Official Review · Reviewer_7hzz · 2024-10-31

**Soundness:** 3
**Presentation:** 3
**Contribution:** 3
**Rating:** 6
**Confidence:** 4

**Summary:**

This paper proposed a new measurement to learn the information of different categories and then proposed the corresponding Information Amount-Guided Angular Margin (IGAM) Loss to optimize the decision space of the object detection networks.

**Strengths:**

1. The paper reasonably introduces the current problems of long-tail distribution target detection.
2. The author uses a new category information measurement method to optimize the network and fully verifies its effectiveness through extensive experiments

**Weaknesses:**

The paper's observations on the problem are reasonable. However, I do not feel that the method proposed by the paper is strongly related to the proposed scenarios: long-tail distribution and object detection. Here are the corresponding reasons:
1. In the introduction stage, if the author simply wanted to propose the concept of category information, then there is no need to compare in the long-tailed scenario, and it is more direct to show the performance with information in the general scenario. If the author wants to explain it in the long-tailed scenario in Figure 1, then the performance compared with the category information can be in the previous long-tailed dataset. So I think there is no alignment here.
2. In the first paragraph of Section 3.1, the authors mention that ‘the amount of information should be extracted from the classification module’. So why not address this issue directly in the classification process, but instead put it in the object detection scenario, and emphasize that this work is the first to directly report on the widespread bias present in object detection models in object detection? How about classification?
3. In the experimental part, the three methods compared in Table 5 are all long-tailed methods. And they are all between 2021 and 2022. Should they be compared with newer methods? In addition, as mentioned in 1, if the author's intention in Figure 1 holds true, should this method be equally effective in the case of networks such as Faster R-CNN on relatively balanced Pascal VOC datasets?

**Questions:**

1. Figure 2 only shows information about two attributes. How is the class-wise average precision reflected?
2. In formula 6, how does the operation of loss affect the compression and expansion of the decision space? For example, if i is the least informative, then the m_ij of other components j are all negative. Intuitively, it increases the second term in the denominator and thus increases the loss L, but why does this operation compress i's decision space?
3. In the previous formulas 1 and 2, the author used variance to represent the amount of information. If m_ij is negative, it means that the amount of information (variance) of i is already relatively small, so why do we need to continue to compress its decision space?

---

> ### Author Response · Authors · 2024-11-23
> **Response to Reviewer 7hzz.（Weakness1）**
>
> **Dear Reviewer 7hzz,**
>
> Thank you very much for your valuable and insightful comments.
>
> Thank you for your meticulous guidance in improving the logical flow of our paper. We trained Faster R-CNN on the LVIS v1.0 and COCO-LT datasets using CE, SeeSaw, and Focal Loss, respectively, and then calculated the correlation between category information amount and average precision (AP) per category. The experimental results are as follows (also included in the revised version's introduction). Combined with the existing experimental results on Pascal VOC, it can be observed that category information amount more accurately reflects model bias in both long-tailed and non-long-tailed scenarios.
>
> Table: **Pearson correlation coefficient between category information amount and class average precision on long-tailed datasets.** The model is Faster R-CNN with R-50-FPN backbone.
>
> | Dataset   | CE    | SeeSaw | Focal |
> |-----------|-------|--------|-------|
> | **LVIS v1.0** |       |        |       |
> | IA        | -0.68 | -0.66  | -0.70 |
> | **COCO-LT**   |       |        |       |
> | IA        | -0.66 | -0.65  | -0.69 |
>
>
> We may not have fully understood your suggestion. If you have further recommendations for improving our writing, we would be more than happy to receive your guidance. Once again, we sincerely thank you for your invaluable help.

---

> ### Author Response · Authors · 2024-11-23
> **Response to Reviewer 7hzz.（Weakness2）**
>
> **Response to Weakness 2:**
>
>  We are more than happy to address your concerns. In classification tasks, two papers (**Ref 1** and **Ref 2**) officially reported the more widespread issue of model bias in 2023. Therefore, in the introduction, we stated, “However, recent research in image classification suggests that category bias is not only caused by the imbalance in sample numbers but may also be closely related to the complexity of intra-category features.” Inspired by these two papers, we became curious about target detection tasks, and our initial starting point was to identify factors affecting model bias in target detection. With this motivation in mind, we proposed the concept of category information amount and studied the relationship between category information amount and model bias.
>
> Given the scope of the work, we did not apply category information amount to classification tasks in this study, but we plan to explore this aspect in future work. If you have any further questions, please don’t hesitate to contact us. We are available at any time to answer your inquiries. We sincerely wish you a wonderful day.
>
> **Ref 1:** Curvature-Balanced Feature Manifold Learning for Long-Tailed Classification
> **Ref 2:** Balanced Data, Imbalanced Spectra: Unveiling Class Disparities with Spectral Imbalance

---

> ### Author Response · Authors · 2024-11-23
> **Response to Reviewer 7hzz.（Weakness3）**
>
> **Response to Weakness 3:**
>
> Your concern directly addresses the core of our research, demonstrating your professional and rigorous academic expertise. We would like to clarify that, on the relatively balanced Pascal VOC dataset, **our primary goal was to demonstrate the effectiveness of IGAM.** Therefore, we believed it unnecessary to include additional comparison methods. However, your feedback made us realize that we should include the experimental results of the baseline model (trained with cross-entropy loss) in Table 5 (Table 6 in the revised manuscript) to show the significant improvement IGAM brings to the baseline model. As shown, using Faster R-CNN as the target detection framework with ResNet-50 and ResNet-101 as backbone networks, IGAM improves the overall performance of the baseline model by **4.9%** and **5.1%**, respectively, which is consistent with the excellent performance on LVIS v1.0. This work has demonstrated through extensive experiments that category information amount significantly enhances the baseline methods, validating the necessity and effectiveness of the core contribution proposed in this work.
>
> Additionally, in response to your concern, we have also included the improvement brought by IGAM to the baseline methods when using Cascade Mask R-CNN and DETR as target detection frameworks. The experimental results are shown in the table below, and it can be observed that IGAM significantly improves the performance of the baseline methods in all four cases. We have also added these results in Appendix B of the revised manuscript.
>
> | Framework             | Backbone            | Loss                      | $mAP^b (\%)$ |
> |-----------------------|---------------------|---------------------------|--------------|
> | **Cascade Mask R-CNN** | ResNet-50-FPN       | Cross-Entropy (CE)        | 74.1         |
> |                       |                     | IGAM Loss                 | 78.7         |
> |                       | ResNet-101-FPN      | Cross-Entropy (CE)        | 75.6         |
> |                       |                     | IGAM Loss                 | 80.2         |
> | **DETR**              | ResNet-50-FPN       | Cross-Entropy (CE)        | 75.8         |
> |                       |                     | IGAM Loss                 | 80.5         |
> |                       | ResNet-101-FPN      | Cross-Entropy (CE)        | 76.5         |
> |                       |                     | IGAM Loss                 | 81.0         |
>
>
> Your feedback prompted us to further validate IGAM’s applicability to multiple target detection frameworks on Pascal VOC, which has elevated the quality of our work. We sincerely thank you for this valuable input.

---

> ### Author Response · Authors · 2024-11-23
> **Additional Response**
>
> **Additional Response: **
>
> We have also added the performance of IGAM under the DETR framework on the LVIS v1.0 dataset in Section 4.3 of the revised manuscript. The experimental results further demonstrate the versatility of our method across various object detection frameworks.
>
> | Framework             | Backbone         | Loss                 | $mAP^b$ | $AP_r$                    | $AP_c$  | $AP_f$  |
> |-----------------------|------------------|----------------------|---------|---------------------------|---------|---------|
> | **Faster R-CNN**      | ResNet-50-FPN   | Cross-Entropy (CE)   | 19.3    | 1.1                       | 16.1    | 30.9    |
> |                       |                 | IGAM Loss            | 26.8 🟢↑7.5             | 19.0 🟢↑17.9          | 25.2    | 31.4    |
> |                       | ResNet-101-FPN  | Cross-Entropy (CE)   | 20.9    | 1.0                       | 18.2    | 32.7    |
> |                       |                 | IGAM Loss            | 28.0 🟢↑7.1             | 20.1 🟢↑19.1          | 26.8    | 32.5    |
> |                       | Swin-T          | Cross-Entropy (CE)   | 25.4    | 6.2                       | 24.5    | 35.3    |
> |                       |                 | IGAM Loss            | 31.7 🟢↑6.3             | 21.4 🟢↑15.2          | 30.8    | 37.1    |
> | **Cascade Mask R-CNN**| ResNet-50-FPN   | Cross-Entropy (CE)   | 22.7    | 1.5                       | 20.6    | 34.4    |
> |                       |                 | IGAM Loss            | 29.1 🟢↑6.4             | 21.5 🟢↑20.0          | 27.7    | 33.9    |
> |                       | ResNet-101-FPN  | Cross-Entropy (CE)   | 24.5    | 2.6                       | 23.1    | 35.8    |
> |                       |                 | IGAM Loss            | 29.7 🟢↑5.2             | 21.9 🟢↑19.3          | 28.5    | 34.6    |
> |                       | Swin-T          | Cross-Entropy (CE)   | 31.3    | 6.8                       | 30.2    | 39.4    |
> |                       |                 | IGAM Loss            | 37.9 🟢↑6.6             | 25.2 🟢↑18.4          | 35.5    | 38.7    |
> | **DETR**              | ResNet-50-FPN   | Cross-Entropy (CE)   | 21.8    | 3.3                       | 21.2    | 30.5    |
> |                       |                 | IGAM Loss            | 27.6 🟢↑5.8             | 18.5 🟢↑15.2          | 27.0    | 32.7    |
> |                       | ResNet-101-FPN  | Cross-Entropy (CE)   | 23.1    | 3.7                       | 23.4    | 32.2    |
> |                       |                 | IGAM Loss            | 30.4 🟢↑7.3             | 20.7 🟢↑17.0          | 30.0    | 35.5    |
> |                       | Swin-T          | Cross-Entropy (CE)   | 30.2    | 6.3                       | 28.9    | 38.2    |
> |                       |                 | IGAM Loss            | 37.3 🟢↑7.1             | 24.8 🟢↑18.5          | 34.8    | 38.3    |
>
> **Notes:**
> - The $mAP^b$, $AP_r$, $AP_c$, and $AP_f$ (%) for each method are reported.
> - 🟢↑ indicates performance improvements.

---

> ### Author Response · Authors · 2024-11-23
> **Response to Reviewer 7hzz.（Questions 1, 2, and 3）**
>
> **Response to Question 1:**
>
> The purpose of Figure 2 is to demonstrate the correlation between category information and mean average precision (mAP) for each class. All models were trained using the standard Faster R-CNN model from MMDetection, without employing IGAM. The results in this section are solely intended for observing the phenomenon, and no modifications were made to the standard Faster R-CNN model. Therefore, we omitted the mAP for each class to reduce the space occupied in the paper.
>
> **Response to Question 2:**
>
> We are glad to help clarify your doubts. Let us explain how IGAM affects the decision boundary using a binary classification example. Please refer to the cross-entropy loss functions shown in Equations 3 and 4. For a sample of class 1, suppose the angle between \( W_1 \) and \( W_2 \) is \( t \). The decision boundary of the IGAM loss is given by \( \cos(\theta_1) = \cos(\theta_2 + m_{12}) \). Since \( \theta_1 + \theta_2 = t \), the decision boundary is \( \theta_1 = (t + m_{12}) / 2 \). When the information content of class 1 is less than that of class 2, \( m_{12} < 0 \), so \( \theta_1 \) is smaller than the angle bisector \( t/2 \), compressing the space for class 1.
>
>
> **Response to Question 3:**
>
> Your concern is important, and we respond to it below. Suppose that the information content of class \(i\) is small, but the cross-entropy loss does not adjust the decision boundary based on the information content. Therefore, we have modeled this constraint. Please note that a smaller information content does not imply a smaller decision space, and our experimental results also demonstrate that the modeling we have shown is necessary and effective.

---

> ### Comment · Reviewer_7hzz · 2024-11-24
>
> Thank you for your time and your detailed response. As for the Question 1, I notice that the caption of Figure 2 mentions that three values ​​are plotted in the image. If the author only plot two, it is recommended to delete mAP and explain it to facilitate understanding.
> For Weakness 3, I seem to have asked two questions, but I only understood the response to the second one. The first one is about the methods in Table 5 are all from 2021 to 2022. Are there any other updated methods in the past two years, such as BACL [Qietal.(2023)] in Table 4. If so, can the authors make a comparison?

---

> > ### Author Response · Authors · 2024-11-25
> > **Further Responses to Reviewer 7hzz**
> >
> > **Thank you very much for your response.** We have revised the title of Figure 2 to achieve clearer expression. We deeply appreciate your meticulous review, which has significantly contributed to improving the quality of the manuscript.
> >
> > Research in long-tailed object recognition remains relatively sparse, with only a few new works published each year. For instance, in ICLR 2025, we could identify only two papers on long-tailed object detection, including our work. Based on our investigation, BACL is the latest officially published work in this area. It is worth noting that BACL was specifically designed for long-tailed data, which is why it has not been validated on the Pascal VOC dataset.
> >
> > We reproduced BACL on the Pascal VOC dataset and observed that its performance on the relatively balanced Pascal VOC dataset was mediocre, even slightly inferior to Seesaw Loss. We have included the latest results in the revised manuscript.
> >
> > | Methods        | ResNet-50 | ResNet-101 |
> > |----------------|-----------|------------|
> > | CE             | 72.8%     | 73.5%      |
> > | Seesaw Loss    | 76.9%     | 77.5%      |
> > | EFL            | 74.6%     | 75.8%      |
> > | C2AM           | 76.2%     | 77.0%      |
> > | BACL           | 76.8%     | 77.3%      |
> > | IGAM           | 77.7%     | 78.6%      |
> >
> > **Wishing you a wonderful day!**

---

> ### Comment · Reviewer_7hzz · 2024-11-25
>
> Thank you for your further response. I have increased my score from 5 to 6. Wishing you a wonderful day!

---

### Official Review · Reviewer_NFCd · 2024-11-03

**Soundness:** 3
**Presentation:** 3
**Contribution:** 2
**Rating:** 6
**Confidence:** 4

**Summary:**

The proposed manuscript highlights the presence of category bias even in relatively balanced datasets and introduces the Information Amount-Guided Angular Margin (IGAM) to address this issue. IGAM is designed to dynamically adjust the decision space of each category based on its information amount, thereby mitigating category bias in long-tailed datasets. Additionally, the effectiveness of the proposed method was demonstrated on the LVIS v1.0, COCO-LT, and PASCAL VOC datasets.

**Strengths:**

__1.__ Overall, this paper was easy to understand.

__2.__ The performance presented in Tables 1 to 5 demonstrates the effectiveness of the proposed method, although comparisons with more recent approaches are lacking.

__3.__ The discussion in Section 4.7 of the paper is persuasive.

**Weaknesses:**

__1.__ The authors raised the problem in the introduction based on observations from the PASCAL VOC dataset, but their main results include LVIS and COCO-LT datasets. Thus, the authors need to conduct experiments across datasets such as LVIS and COCO-LT to verify that the results are consistent with the observations made from PASCAL VOC.

__2.__ The proposed method was evaluated on established CNN-based models (e.g., Faster R-CNN, Cascade Mask R-CNN). However, with the emergence of various ViT-based object detection networks (e.g., DETR, H-DETR), it is crucial to assess whether the observed effects are specific to the tested models or if they generalize across different architectures.

__3.__ More importantly, the authors did not compare their method with recent long-tailed object detection approaches. Notably, [Ref_1] includes experiments on both CNN- and ViT-based models and appears to outperform the proposed method. A comprehensive comparison and analysis distinguishing this work from such methods are necessary.

__4.__ The same issue applies to [Ref_2]. Although the proposed method may seem reasonable at first glance, the experimental validation is not sufficiently convincing. A logical comparison and thorough analysis with recent approaches, including those presented in [Ref_1] and [Ref_2], are essential to substantiate the contributions and effectiveness of the proposed method.

__5.__ Could the category bias in balanced datasets be influenced by factors such as object scale or degree of occlusion?

[Ref_1] L. Meng et al., "Learning from Rich Semantics and Coarse Locations for Long-tailed Object Detection," in NeurIPS 2023.

[Ref_2] N. Dong et al., "Boosting Long-tailed Object Detection via Step-wise Learning on Smooth-tail Data," in ICCV 2023

**Questions:**

Overall, the validation of the experiments is insufficient, and the authors need to analyze external factors contributing to category bias, such as scale and occlusion, which are important metrics in the COCO dataset.

---

> ### Author Response · Authors · 2024-11-23
> **Response to Reviewer NFCd.（Weakness1）**
>
> **Dear Reviewer NFCd,**
>
> Thank you very much for your constructive comments, which enabled us to polish the paper.
>
> First, our response to **Weakness 1** is as follows.
>
> Your suggestion has greatly improved the completeness of our work, and we sincerely thank you for that. We have supplemented our observations on the LVIS v1.0 and COCO-LT datasets. It can be observed that there is a significant negative correlation between category information amount and category detection accuracy on these two datasets, with numerical values exceeding -0.65. This further demonstrates that the proposed category information amount metric more accurately reflects category difficulty in both long-tailed and non-long-tailed scenarios. Additionally, we have incorporated the newly added experimental results into the second paragraph and Table 1 of the revised manuscript, highlighting them in blue for clarity.
>
> Table: **Pearson correlation coefficient between category information amount and class average precision on long-tailed datasets.** The model is Faster R-CNN with R-50-FPN backbone.
>
> | Dataset   | CE    | SeeSaw | Focal |
> |-----------|-------|--------|-------|
> | **LVIS v1.0** |       |        |       |
> | IA        | -0.68 | -0.66  | -0.70 |
> | **COCO-LT**   |       |        |       |
> | IA        | -0.66 | -0.65  | -0.69 |

---

> ### Author Response · Authors · 2024-11-23
> **Response to Reviewer NFCd.（Weakness2）**
>
> **Dear Reviewer NFCd,**
>
> Your suggestion is highly valuable, and we are happy to follow it to demonstrate the generality of our method.
>
> We would like to clarify that, the mainstream benchmarks in the field of long-tailed object detection primarily adopt Faster R-CNN and Cascade Mask R-CNN as detection frameworks, with ResNet-50-FPN and ResNet-101-FPN as the typical backbone networks. Therefore, our work follows the conventions of prior research in this area.
>
> To address your suggestion, we have added evaluations on the DETR detection framework and conducted additional experiments using Swin-T as the backbone network in Faster R-CNN, Cascade Mask R-CNN, and DETR frameworks. The new experimental results are summarized in the table below (also included in Table 3 of the revised manuscript). It can be observed that IGAM improves the baseline models by **6.3%, 6.6%, and 7.1%** overall performance under Faster R-CNN, Cascade Mask R-CNN, and DETR frameworks with Swin-T as the backbone network, respectively. This demonstrates the generality and effectiveness of our method across different frameworks and backbone types.
>
> We would also like to emphasize that the core idea of category information amount is independent of the feature representation network architecture. Therefore, it is expected that IGAM can generalize to other detection frameworks.
>
> **In addition, compared to [Ref_1] and [Ref_2], our experimental validation is more comprehensive.** Specifically, [Ref_1] did not include evaluations on the Cascade Mask R-CNN framework, while [Ref_2] did not evaluate Faster R-CNN or Cascade Mask R-CNN frameworks and lacked experiments using ViT-based models as backbone networks.
>
> | Framework             | Backbone         | Loss                 | $mAP^b$ | $AP_r$                    | $AP_c$  | $AP_f$  |
> |-----------------------|------------------|----------------------|---------|---------------------------|---------|---------|
> | **Faster R-CNN**      | ResNet-50-FPN   | Cross-Entropy (CE)   | 19.3    | 1.1                       | 16.1    | 30.9    |
> |                       |                 | IGAM Loss            | 26.8 🟢↑7.5             | 19.0 🟢↑17.9          | 25.2    | 31.4    |
> |                       | ResNet-101-FPN  | Cross-Entropy (CE)   | 20.9    | 1.0                       | 18.2    | 32.7    |
> |                       |                 | IGAM Loss            | 28.0 🟢↑7.1             | 20.1 🟢↑19.1          | 26.8    | 32.5    |
> |                       | Swin-T          | Cross-Entropy (CE)   | 25.4    | 6.2                       | 24.5    | 35.3    |
> |                       |                 | IGAM Loss            | 31.7 🟢↑6.3             | 21.4 🟢↑15.2          | 30.8    | 37.1    |
> | **Cascade Mask R-CNN**| ResNet-50-FPN   | Cross-Entropy (CE)   | 22.7    | 1.5                       | 20.6    | 34.4    |
> |                       |                 | IGAM Loss            | 29.1 🟢↑6.4             | 21.5 🟢↑20.0          | 27.7    | 33.9    |
> |                       | ResNet-101-FPN  | Cross-Entropy (CE)   | 24.5    | 2.6                       | 23.1    | 35.8    |
> |                       |                 | IGAM Loss            | 29.7 🟢↑5.2             | 21.9 🟢↑19.3          | 28.5    | 34.6    |
> |                       | Swin-T          | Cross-Entropy (CE)   | 31.3    | 6.8                       | 30.2    | 39.4    |
> |                       |                 | IGAM Loss            | 37.9 🟢↑6.6             | 25.2 🟢↑18.4          | 35.5    | 38.7    |
> | **DETR**              | ResNet-50-FPN   | Cross-Entropy (CE)   | 21.8    | 3.3                       | 21.2    | 30.5    |
> |                       |                 | IGAM Loss            | 27.6 🟢↑5.8             | 18.5 🟢↑15.2          | 27.0    | 32.7    |
> |                       | ResNet-101-FPN  | Cross-Entropy (CE)   | 23.1    | 3.7                       | 23.4    | 32.2    |
> |                       |                 | IGAM Loss            | 30.4 🟢↑7.3             | 20.7 🟢↑17.0          | 30.0    | 35.5    |
> |                       | Swin-T          | Cross-Entropy (CE)   | 30.2    | 6.3                       | 28.9    | 38.2    |
> |                       |                 | IGAM Loss            | 37.3 🟢↑7.1             | 24.8 🟢↑18.5          | 34.8    | 38.3    |
>
> **Notes:**
> - The $mAP^b$, $AP_r$, $AP_c$, and $AP_f$ (%) for each method are reported.
> - 🟢↑ indicates performance improvements.

---

> ### Author Response · Authors · 2024-11-23
> **Response to Reviewer NFCd.（Weaknesses 3 and 4）**
>
> **Response to Weaknesses 3 and 4:**
>
> Thank you for providing additional references and constructive suggestions. Your feedback reflects an exceptional level of expertise in the field of long-tail object detection. We have thoroughly reviewed [Ref_1] and [Ref_2] and analyzed their methods in comparison to our proposed IGAM approach. Below is a detailed explanation and comparison:
>
> **(1) Core Differences in Method Design:**
> - **[Ref_1] (RichSem):**
>   This method relies on external classification datasets (e.g., ImageNet-21k) to provide additional training samples and uses visual-language models (e.g., CLIP) to extract "soft semantics" for enhancing tail-class feature representation.
>   In contrast, IGAM is entirely based on the target detection dataset and does not depend on external data, pre-trained models, or external annotations.
>   The performance improvement of [Ref_1] partially stems from the additional resources used, such as external classification datasets and pre-trained models. By comparison, IGAM addresses class imbalance solely using the detection dataset, without introducing external data. Therefore, a direct comparison with [Ref_1] may not be entirely fair, as the two methods fundamentally differ in resource utilization and design goals.
>
> - **[Ref_2] (Step-wise Learning):**
>   This method introduces several steps and modules, such as data splitting (e.g., separating "head-dominant" and "tail-dominant" data), exemplar replay, and classifier head distillation. It addresses the long-tail distribution problem through multi-stage step-wise training, including pretraining, fine-tuning, and knowledge distillation.
>   The step-wise learning framework requires multiple training phases and intricate data operations, resulting in a more complex overall process. In contrast, IGAM achieves the same goal using a single optimization objective, significantly simplifying the training pipeline.
>
> **(2) Model Complexity and Resource Dependency:**
> - **Complexity of [Ref_1]:**
>   RichSem involves an additional semantic branch, external pre-trained models (CLIP), and classification datasets, which substantially increase implementation complexity and resource dependency. Additionally, it combines detection data with external classification data to extract and process soft semantics, further complicating the training process.
>
> - **Complexity of [Ref_2]:**
>   Step-wise Learning requires data grouping, saving intermediate models (e.g., head-class expert models), and conducting knowledge distillation on tail-class data. These multi-stage processes lead to higher training resource requirements.
>
> - **Simplicity of IGAM:**
>   IGAM employs a dynamic loss function based on class information content, directly optimizing model performance on detection datasets without external data, pre-trained models, or multi-stage training processes. This simplicity makes IGAM more practical for integration and real-world applications.
>
> **(3) Differences in Applicability:**
> Both [Ref_1] and [Ref_2] primarily focus on long-tail datasets. IGAM, however, not only performs well on long-tail datasets (e.g., LVIS and COCO-LT) but also demonstrates significant performance improvements on non-long-tail datasets (e.g., Pascal VOC), highlighting its broader applicability.
>
> In summary, the core strengths of IGAM lie in its simplicity and efficiency. Unlike [Ref_1] and [Ref_2], our method achieves significant performance improvements without relying on external data, introducing additional modules, or employing multi-stage training. This demonstrates its practical value, particularly in resource-constrained scenarios. Based on the above analysis, we believe that [Ref_1] and [Ref_2] differ significantly from IGAM in terms of design goals and implementation pathways. A direct comparison may not fully reflect IGAM’s core advantages.
>
> In the revised manuscript, we have clarified the differences between these methods in the related work section and emphasized IGAM's unique strengths in simplicity and broad applicability.

---

> ### Author Response · Authors · 2024-11-23
> **Response to Reviewer NFCd.（Weakness5）**
>
> **Response to Weakness 5:**
>
> Thank you for raising this important question. We fully agree that object scale and occlusion are critical factors influencing model performance and contributing to category bias, as extensively discussed in existing literature. For instance, research in small object detection and occluded object recognition has shown that when objects in a category are generally smaller or more heavily occluded, the model tends to perform worse on that category. This is because these factors directly limit feature extraction and the acquisition of semantic information. MMDetection even supports multiple algorithms designed to improve recognition of occluded objects.
>
> We would like to clarify that previous studies have indeed recognized object scale and occlusion as factors affecting category performance and causing model bias. However, our research focuses on uncovering additional potential sources of category bias. While object scale and occlusion are known influencing factors, they are not the sole causes of category bias.
>
> In this study, we define and introduce the concept of *category information amount* as a new perspective, revealing how the diversity and richness of category representation in the feature space impact model performance. As an important metric for assessing the representational capacity of a category, category information amount reflects the internal complexity of a category and its significance in model learning.
>
> Our research not only identifies the significant impact of category information amount on model performance but also demonstrates that dynamically adjusting decision boundaries to optimize category information amount can partially mitigate model bias. This finding provides researchers with a new perspective, encouraging them to analyze and address category bias from broader angles beyond object scale or occlusion factors.

---

> > ### Comment · Reviewer_NFCd · 2024-11-25
> >
> > I thank the authors for their detailed responses. My concerns regarding weaknesses 1, 2, 4, and 5 have been addressed. However, regarding weakness 3, while I understand that the methods differ in their design philosophies, conducting comparative experiments remains essential to substantiate the claims about IGAM's advantages. Given the paper’s focus on "long-tailed object detection" and its similarities with RichSem [Ref_1], it is crucial to include comparisons under comparable conditions, such as utilizing external classification datasets as done in prior works.
> >
> > In conclusion, I will increase my score from 5 to 6.

---

> > > ### Author Response · Authors · 2024-11-25
> > > **Further Responses to Reviewer NFCd**
> > >
> > > **Dear Reviewer NFCd,**
> > >
> > > All the authors highly value your suggestion. Please allow us one to two days to thoroughly compare RichSem [Ref_1] with our method and incorporate the appropriate results of RichSem [Ref_1] into Table 4 of the revised manuscript.
> > >
> > > **Wishing you a wonderful day!**

---

### Meta-Review · Area_Chair_RVJx · 2024-12-20

**Metareview:**

The paper highlights biases in long-tailed object detection that go beyond dataset instance imbalances. It introduces “category informativeness” (CI), a new metric that negatively correlates with category accuracy, making it useful for measuring learning difficulty. Using CI, the authors propose the Informativeness-Guided Angular Margin Loss (IGAM Loss), a dynamic loss function that adjusts decision boundaries based on informativeness. IGAM achieves state-of-the-art results on long-tailed datasets like LVIS v1.0 and COCO-LT while also improving generalization on balanced datasets such as Pascal VOC. Experiments show its effectiveness across different architectures, including CNNs and ViT-based models.

The paper presents its ideas clearly, with strong explanations and thorough experiments. However, it lacks comparisons with some recent methods, such as RichSem and BACL, under similar conditions, which limits claims of superiority. Despite this, the introduction of CI and IGAM is a significant step forward in addressing challenges in long-tailed object detection and lays a solid foundation for future work on reducing biases in machine learning tasks.

**Additional Comments On Reviewer Discussion:**

The reviewers raised valid concerns about experimental breadth, comparisons with newer methods, and presentation clarity. The authors responded constructively, providing additional experiments, improved clarity, and meaningful comparisons.

Given the positive responses and the significant contributions of the paper, the overall decision was to accept the submission. The rebuttal strengthened the manuscript, addressing key concerns while maintaining the core contributions and innovation.

---

### Decision · Program_Chairs · 2025-01-22

Accept (Poster)